# Sorption of Cu$^{2+}$ Ions by Bentonite Modified with Al Keggin Cations and Humic Acid in Solutions with pH 4.5

**Yulia Izosimova [1], Inna Tolpeshta [1], Irina Gurova [1], Michail Karpukhin [1], Sergey Zakusin [2,3] and Victoria Krupskaya [2,4,*]** 

[1]  Soil Science Faculty, M.V. Lomonosov Moscow State University, 119991 Moscow, Russia; izosimova.julya@yandex.ru (Y.I.); itolp@soil.msu.ru (I.T.); gurovairene@gmail.com (I.G.); kmm82@yandex.ru (M.K.)
[2]  Geological Faculty, M.V. Lomonosov Moscow State University, 119991 Moscow, Russia; zakusinsergey@gmail.com
[3]  Institute of Ore Geology, Petrography, Mineralogy and Geochemistry, Russian Academy of Science (IGEM RAS), 119017 Moscow, Russia
[4]  Nuclear Safety Institute, Russian Academy of Science (IBRAE RAS), 115191 Moscow, Russia
*   Correspondence: krupskaya@ruclay.com; Tel.: +7-819-6398

**Abstract:** The sorption of Cu$^{2+}$ onto bentonite modified with Al Keggin cations and humic acid from CuCl$_2$ solutions at pH 4.5 was studied. Modification of Na-bentonite with Al Keggin cations was found to result in an increase in the basal spacing of montmorillonite from 1.29 nm for N-form to 1.85 and 1.78 nm for HAl$_{13}$ and Al$_{13}$ forms respectively, in a reduction of CEC (cation exchange capacity) and in the formation of additional sites with a variable charge with pH$_{PZC}$ 4.2. Al$_{13}$-bentonite is not affected by heat. Under the conditions of the experiments at pH of 4.5 Na-bentonite adsorbs more Cu$^{2+}$ from CuCl$_2$ solutions then Al$_{13}$ forms of bentonites. The main mechanism of copper sorption on Na-bentonite is the cation exchange Cu$^{2+}$–Na$^+$. The reduction of CEC of Na-bentonite after modification with Al Keggin cations leads to a decrease in the Cu$^{2+}$ sorption. pH-dependent sorption sites on Al$_{13}$-bentonites have a pH$_{PZC}$ of 4.2 and, therefore, under conditions of the experiment have positive charge which prevents Cu$^{2+}$ sorption. Na-bentonite adsorbs more humic acid solution (HA) then Al$_{13}$-bentonite and the proportion of adsorbed HA remains constant over the entire concentration range. Treatment of the Al$_{13}$-bentonite with HA leads to the formation of the additional sorption sites. The amount of sorbed Cu$^{2+}$ and the percentage of their extraction from solutions by HAAl$_{13}$-bentonite is similar to those values for Na-bentonite.

**Keywords:** bentonite; pillared smectite; sorption of heavy metals

## 1. Introduction

Natural bentonite clays are widely used as sorbents for heavy metals and radionuclides due to their high sorption capacity and relatively low price [1].

The sorption capacity of bentonite clays towards these pollutants depends on their composition and varies in clays from different deposits [2–12]. It was demonstrated that depending on the pH, ionic strength, composition of the solution and the conditions of the experiments, bentonites with different mineral composition and physical properties, adsorb from 13.2 to 32.7 mg Cu$^{2+}$ per gram clay [13–16]. The main mechanisms of sorption of Cu(II) ions on bentonite are ion exchange and proton substitution of aluminol and silanol groups on the edge surfaces of clay minerals. Ion exchange is independent of pH. Sorption on aluminol and silanol groups depends on pH.

A number of modification methods are used to improve the sorption capacity of clays. This article discusses two methods of bentonite modification: saturation with Al Keggin cations and treatment with humic acid (HA). The saturation of clays with Al Keggin cations followed by calcination results in strong heat-resistant pillared structures. These kinds of structures are characterized by decreased sorption capacity and increased specific surface area and total pore volume [17,18]. In the literature, there are conflicting reports on the sorption capacity of bentonites which have been modified with Al Keggin cations ($Al_{13}$-bentonite). It has been shown that at pH 4.9 $Al_{13}$-bentonite adsorbed less $Cu^{2+}$ ions, compared with the initial Na-bentonite [19]. However, other authors reported that $Al_{13}$-bentonite could adsorb a higher amount of $Cu^{2+}$ ions compared to Na-bentonite in the pH range from 3 to 6 [20]. These latter authors demonstrated that the amount of adsorbed $Cu^{2+}$ ions increases with an increase of density of "pillars" in the interlayer space and $Cu^{2+}$ ions are sorbed both by cationic exchange and by forming complexes on the surface of the aluminum oxide "pillars".

Humic acid treatment of smectite leads to an increase in the sorption capacity towards $Cr^{3+}$, $Cu^{2+}$ and $Cd^{2+}$ [21]. After modification with Keggin cations and calcination bentonite becomes stable to heating and its swelling reduces, which makes it possible to use it as an effective sorbent and catalyst under certain conditions. However, the modification causes the decrease in CEC. The sorptive capacity of the Al13-bentonite can be improved by treating its surface with humic acid.

In an acidic environment, the sorption of cations on the surface of bentonite treated with humic acid will be limited by the pKa values of the carboxyl groups of humic acid, which are known to vary in the range 5–7.5. It is necessary to establish the efficiency of bentonite modified in this way for the sorption of cations in the acidic environment.

Since the sorption capacity of bentonites modified by Al Keggin cations and HA depend on their properties and the sorption conditions, the well-known sorption mechanisms do not always explain the experimental results that can sometimes be contradictory, further research is needed in this field.

The aim of this research is to reveal the regularities of the sorption of $Cu^{2+}$ by bentonite modified with Al Keggin cations and humic acid from $CuCl_2$ solutions at pH 4.5.

## 2. Materials and Methods

A <1 μm clay fraction from a bentonite sample of the Sarigukhskoie deposit (Tavush region, Armenia) was used for the research. It was isolated by sedimentation after the removal of carbonates from the clay with 10% HCl solution. The obtained sample of clay fraction will be referred to as "bentonite" hereinafter.

The pH of the clay fraction suspension in Ca and Na forms were measured at a clay: water ratio of 1:1000 and was estimated at 6.21 and 6.38 units respectively. The clay-sized fraction of the studied bentonite (particle size < 1 μm) strongly swells, which prevents the proper mixing of the suspension. Good mixing is especially important when measuring the pHzpc value by titration procedure. In a preliminary experiment dilutions 1:500, 1:750 and 1:1000 were tested, and the best titration curves were obtained at the dilution of 1:1000. This is why the clay:solution ratio of 1:1000 (0.02 g:20 mL) was used. This ratio is often applied when the zeta potential is measuring [22,23].

### 2.1. Preparation of the Al-Pillared Bentonite

The modification of the silty fraction of bentonite was done according to the article [24] from Na-smectite. The solution of Al Keggin cations $[AlO_4Al_{12}(OH)_{24}(H_2O)_{12}]^{7+}$ was prepared by alkaline hydrolysis of 0.1 M $AlCl_3$ solution by titration with 1 M NaOH on a magnetic stirrer to the molar ratio OH:Al = 2.4. The obtained solution was aged for 7 days at room temperature.

Na-smectite was prepared from the Ca-form by repeated treatment with 1 M NaCl solution. After saturation, the sample was washed from the $Cl^-$ ions by dialysis and checked by reaction with $AgNO_3$. 1% aqueous suspension of Na-bentonite was prepared by mixing on a magnetic stirrer for 5 h at 20 °C, and then left to stand overnight at room temperature. Al Keggin cations solution was added to the Na-bentonite suspension dropwise at a constant stirring on a magnetic mixer at 22 °C.

The obtained solution was kept at room temperature for 24 h and then centrifuged. Smectite saturated with polyhydroxy-aluminum ($HAl_{13}$-smectite), was washed with distilled water to remove $Cl^-$ ions and then dried at room temperature. The removal of hydroxyl groups from the Al Keggin cations was performed by calcination of the $HAl_{13}$-bentonite at 400 °C for 4 h. The calcined $HAl_{13}$-bentonite form hereinafter will be denoted as $Al_{13}$-bentonite. Saturation with polyhydroxy-aluminum decreased the pH of the clay suspension compared to the Ca and Na forms to the values 5.38 and 5.45 for the $HAl_{13}$- and $Al_{13}$-bentonites respectively.

Experiments on the sorption of $Cu^{2+}$ on the Na-, $HAl_{13}$-, and $Al_{13}$-bentonites. The first experiment studied the dependence of the clay sorption capacity towards $Cu^{2+}$ on the reaction time with 0.5 mM $CuCl_2$ solution at pH 4.5, which was maintained by adding 0.1 N HCl solution. Experiments were carried out at a solid:liquid ratio of 1:1000. Suspensions were shaken at 250 rpm for 5, 10, 20, 30, 40 and 60 min. After that, the pH was measured and the suspension was centrifuged for 5 min at 5000 rpm. Then the supernatant was passed through the 0.45 μm filter and the Cu(II) concentration was determined.

The second experiment studied the dependence of the Na- and Al-bentonites sorption capacity towards $Cu^{2+}$ in $CuCl_2$ solutions. Experiments were carried out at a solid:liquid ratio of 1:1000 where the concentrations of Cu(II) were 0.025, 0.05, 0.125, 0.250 and 0.5 mM, all at pH 4.5. The ionic strength of the resultant copper chloride solutions ranged from 7.5.10-5 to 1.5.10-3 M. The suspensions were shaken for 30 min at 250 rpm. After measuring the pH, the suspension was centrifuged for 5 min at 5000 rpm. Then the supernatant was passed through a 0.45 μm filter and the Cu(II) concentration was measured.

Based on the data obtained it was established that the maximum sorption of $Cu^{2+}$ is achieved within 30 min. Thus, this time interval was chosen as standard for the main sorption experiments.

## 2.2. Experiments on the Sorption of Humic Acid by Na- and $Al_{13}$-bentonite

A humic acid solution (HA) was prepared from a dry sample, produced by the company "Biolar". A sample with a mass of 0.735 g was poured into 500 mL of distilled water, then 0.75 mL of 1 M NaOH was added dropwise and the whole volume was adjusted to 1000 mL with distilled water. The obtained solution was filtered through a White Ribbon filter and the Corg concentration was determined. The dependencies of the HA sorption by the different bentonite forms on different factors were studied. The experiments on the effect of the sorbent content were carried out at clay:solution ratios of 1:1000, 1:750 and 1:500 at constant pH of 4.5. The experiments were carried out at an ionic strength of 0.1 M, which was maintained using NaCl solution; the interaction time was 30 min. The ionic strength is actually high but such values of ionic strength is commonly applied in similar kinds of experiments [25–27].

The dependence of the HA sorption on the time of interaction was studied using solution with concentration Corg 24.35 mM at pH 4.5 with a clay:solution ratio of 1:750 and ionic strength of 0.1 M. The suspensions were shaken at 200 rpm for 0.25, 0.5, 1, 2, 4, 6 and 8 h. Then the supernatants were passed through a White Ribbon filter. The HA concentrations were determined by the photometric method.

The dependence of the HA sorption on pH was studied using 24.35 mM and 6.09 M solutions at pH values of 3.0, 3.5, 4.5 and 7.0 at a constant ionic strength of 0.1 M and a solid:liquid ratio—1:750 as was chosen according to the preliminary experiments. It was found that the highest sorption by the $Al_{13}$-smectite is achieved at pH 3 so it was chosen for the following experiment.

To determine the sorption dependence on the initial concentration of humic acid the following concentrations were used: 6.09, 12.18, 18.26, 24.35 mM. The experiments were carried out at pH 3 with a clay:solution ratio of 1:750 and ionic strength of 0.1 M. In both experiments, the suspensions were shaken for 6 h at 200 rpm. Then the supernatants were filtered through a White Ribbon filter and the concentrations of Corg were determined.

The concentration of the Corg solution, at which the adsorption of HA on the $Al_{13}$-bentonite was the highest turned out to be 12.18 mM and was used for the preparation of the HA-$Al_{13}$-bentonite for the following experiment.

## 2.3. Experiment on the Sorption of the $Cu^{2+}$ on the HA-$Al_{13}$-Bentonites

$Al_{13}$-bentonite was saturated with HA (HA-$Al_{13}$) in a solution with a Corg concentration of 12.18 mM, I = 0.1 M, pH 3 and at a clay:solution ratio of 1:750 for 6 h. The experiment was conducted under the same conditions as the experiments on $Cu^{2+}$ sorption by the Na-, $HAl_{13}$- and $Al_{13}$-bentonite forms.

Sorption isotherms of $Cu^{2+}$ and HA ions were described by the Freundlich equation:

$$\lg Q \ = \ \lg k + n \lg C_{eq}$$

where $Q$ is the amount of sorbed $Cu^{2+}$, in mmol/g; $C_{eq}$ is the equilibrium concentration of Cu(II), in mmol/L; and $k$ and $n$ are empirical coefficients.

All the experiments were carried out in duplicate.

## 2.4. Measurements

The raw and modified smectites were studied by X-ray diffraction analysis. Oriented specimens were analyzed with the DRON-3 diffractometer (Burevestnik, St-Petersburg, Russia) with Ni-filtered Cu-K$\alpha$ radiation, with range 2–62°2$\theta$, step size—0.05°2$\theta$, exposure time—10 s. Measurements were carried out for the specimens under the four following conditions: (1) in the air-dry state; (2) ethylene glycol solvated; and calcined at (3) 350 °C and (4) 550 °C, both for 2 h.

The potentiometric method was used to determine the pH of the suspensions and solutions, and for the determination of the pH of the point of zero charge ($pH_{PZC}$). The potentiometric analysis was carried out with the Mettler Toledo Sevengo Pro (Mettler Toledo Inc., New York, NY, USA) and autotitrator Mettler Toledo DL 58 (Mettler-Toledo AG, Schwerzenbach, Switzerland).

The concentration of Cu(II) was measured by atomic absorption on the spectrophotometer ContrAA 300 (Analytik Jena spectrometer, Jena, Germany).

The content of Corg in the initial HA solution was determined by the Tyurin method with the photometric ending at 590 nm wavelength [28]. The Corg concentration in the equilibrium solutions was determined by spectrophotometry at 350 nm wavelength using a UNICO 1201 spectrophotometer (United Products & Instruments Inc., Dayton, NJ, USA), since the absorption maxima at this wavelength correlated well with the DOC concentration.

## 3. Results

### 3.1. Mineral Composition and Properties of Various Bentonite Forms

On the XRD pattern of Ca-bentonite, a series of peaks with d/$n$ values of 12.9 (001), 6.1 (002), 4.04 (003), and 3.13 (004) Å was observed (Figure 1). After saturation with ethylene glycol (EG), the interplanar distance increases to 16.9 Å, and after calcination, it decreases to 9.9Å. It can be concluded that the <1 μm fraction consists almost entirely of smectite. The smaller d(001) values of the Ca-montmorillonite in the air-dry state than what has been described in the literature [29,30] can be explained by a slight modification of the montmorillonite's crystal lattice after the treatment with the 10% HCl solution [5,6,31].

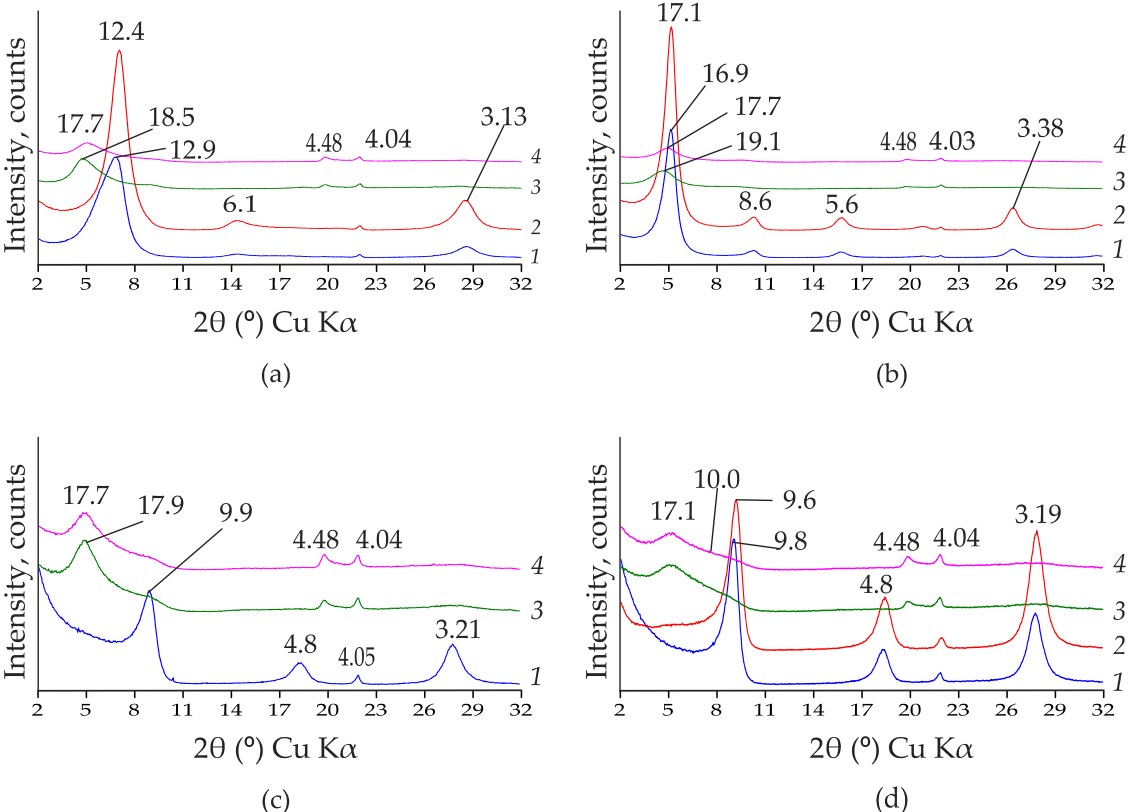

**Figure 1.** XRD patterns of the Ca-, Na-, HAl$_{13}$- and Al$_{13}$-bentonite, <1 μm fraction (1, 2, 3 and 4, respectively) in the air-dry state (**a**), ethylene glycol (EG)-solvated (**b**), calcined at 350 °C (**c**) and 550 °C (**d**). The d(001)-values are given in Å.

The saturation of the smectites with Na$^+$ ions leads to a decrease of the d(001) value of montmorillonite to 12.4 Å while the intercalation with polyhydroxy-aluminum cations increases it up to 18.5 Å. Calcination of HAl$_{13}$-bentonite at 400 °C causes compaction of the lattice along the c-axis which is observed by the d(001) value of 17.8 Å. The decrease in the interplanar distance after the calcination is caused by the reactions of dehydroxylation and the formation of aluminum oxide, which has a smaller size, compared to the Al Keggin cations. Considering the facts that the thickness of the montmorillonite layer is of about 9.6 Å [32], and the diameter of Cu(H$_2$O)$_6$$^{2+}$ is about 5.4 Å [18], the obtained HAl$_{13}$ and Al$_{13}$-smectites with interlayer distances of 8.9 Å and 8.2 Å respectively, will be available for intercalation with Cu$^{2+}$ ions.

After saturation with ethylene glycol, the interplanar distances of smectites in Ca- and Na-bentonite increase by 3.9 Å and 4.7 Å respectively. Montmorillonite in HAl$_{13}$-bentonite swells to a lesser extent and its interplanar distance is increased by only 0.6 Å. The interplanar distance of montmorillonite in Al$_{13}$-bentonite after saturation with ethylene glycol does not change. After calcination at 350 °C and 550 °C, the interplanar distances of montmorillonite in the Ca- and Na-bentonites reduces to 9.6–9.8 Å, whereas Al$_{13}$-montmorillonite, even after calcination at 550 °C, retains a broad peak with d(001) = 17.1 Å (Figure 1).

From the data obtained, it can be concluded that montmorillonite in the Al$_{13}$-bentonite sample has a stable crystal structure and its interplanar spacings were not changed either after saturation with ethylene glycol or after calcination at of 550 °C.

### 3.2. Kinetics of Cu$^{2+}$ Adsorption

In the investigated time interval, Na-bentonite adsorbs more than 2 times as much Cu$^{2+}$ ions compared with the Al forms of bentonite and after 5 min of interaction adsorbs 50–60% of the Cu$^{2+}$

from the 0.5 mM CuCl$_2$ solution with pH 4.5 (Figure 2a,b). Further increasing the interaction time has no significant effect on the sorption of copper. In the range from 10 to 60 min, about 70% or 0.3 mmol/g (19.1 mg/g) Cu$^{2+}$ is sorbed. The obtained values of the maximum amount of sorbed Cu$^{2+}$ ions agrees well with those described in the literature (13–33 mg/g) for unmodified bentonites [13–16,21].

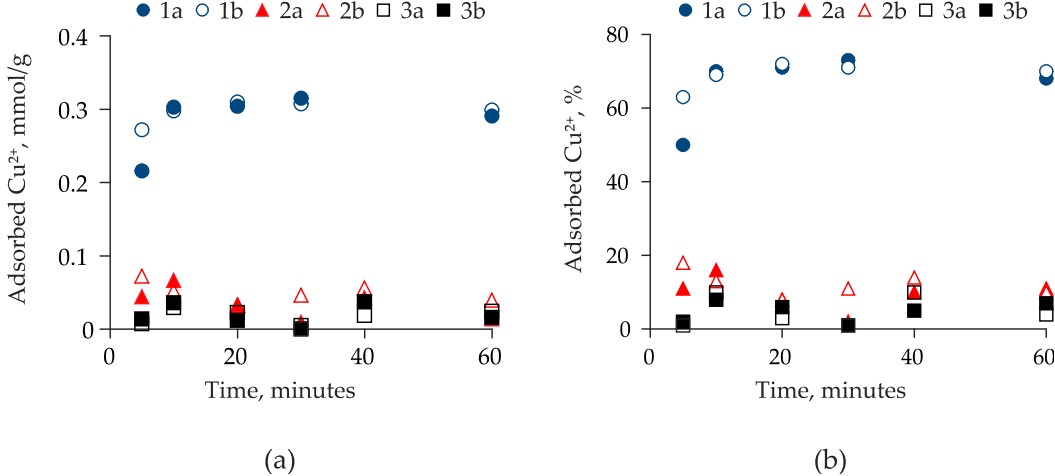

(a)                                                                  (b)

**Figure 2.** The dependence of the sorption of Cu$^{2+}$ in mmol/g (**a**) and % (**b**) on the interaction time. 1, 2, 3 = Na-, HAl$_{13}$- and Al$_{13}$-bentonite, respectively. Empty and filled icons (a,b) = repetitions of the same experiment.

The sorption of copper by the Al forms of bentonite in the entire time interval does not exceed 0.07 mmol/g. The non-calcined bentonite sample extracts 2–18%, while the calcined bentonite sample extracts 1–10%, of the Cu$^{2+}$ from a copper chloride solution.

As a result of the interaction of Na- and HAl$_{13}$-bentonite with the 0.5 mM solution of CuCl$_2$, the equilibrium pH of the suspension increased, compared to the initial pH of copper chloride solution by 0.1 and 0.3 average, respectively. This increase turned out to be independent of the interaction time (Figure 3). The increase of pH can be associated with the neutralizing effect of clay, the pH values of the suspension of which, as already mentioned above, are 6.39 and 5.45 for the Na and HAl$_{13}$ forms respectively.

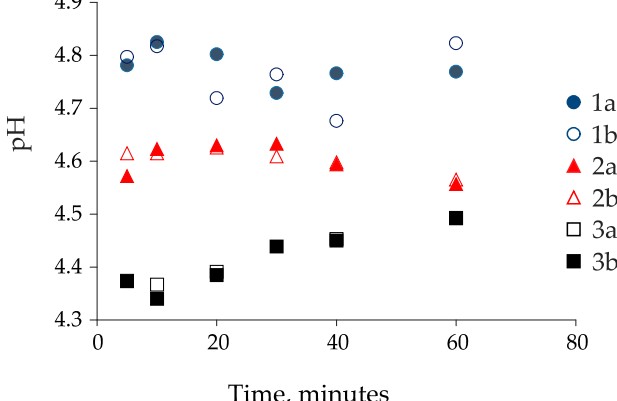

**Figure 3.** The dependence of the pH of the suspension on the time of the experiment. 1, 2, 3 = Na-, HAl$_{13}$- and Al$_{13}$-bentonite, respectively. Empty and filled icons (a,b) = repetitions of the experiment.

In the first 30 min of the interaction with Al$_{13}$-bentonite, the pH decreases in comparison with the initial value of the copper chloride solution. After 5 min of interaction, the pH decreases by 0.1 and after 30 min it returns almost to the initial value of the copper chloride solution.

### 3.3. Sorption of Cu$^{2+}$ Depending on the Concentration of the Initial Solution

In the range of initial concentrations of copper chloride solution, Na-bentonite adsorbs rather more Cu$^{2+}$ ions compared to the Al forms (Figure 4a,b). With an increase in the concentration of copper chloride, the amount of sorbed copper on Na-bentonite increases significantly. This dependence was less significant for the Al-bentonites (Figure 4a).

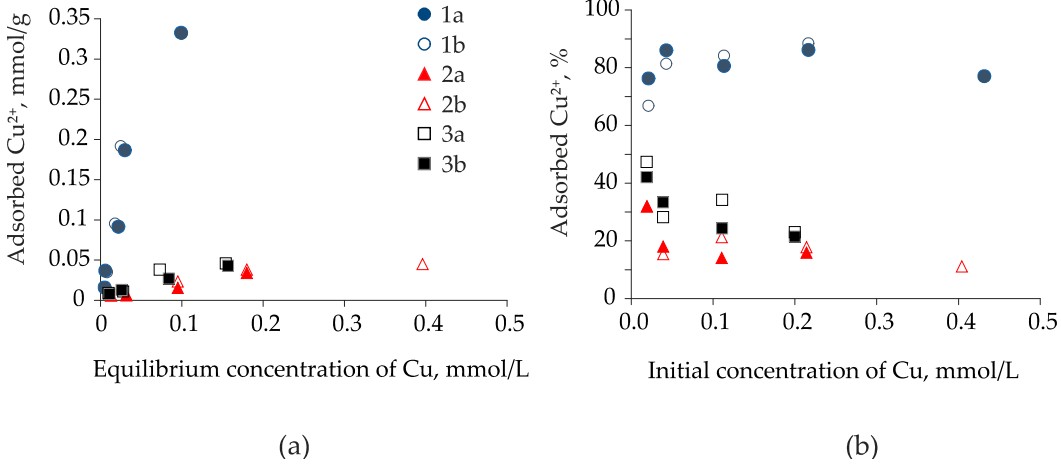

(a)                                                    (b)

**Figure 4.** The dependence of the sorption of Cu(II) in mmol/g (**a**) and % (**b**) on the concentration of the initial solution. 1, 2, 3 = Na-, HAl$_{13}$- and Al13-bentonite, respectively. Empty and filled icons (a,b) = repetitions of the same experiment.

In the whole range of the concentrations tested, Na-bentonite adsorbs about 80% of the Cu(II) ions of the initial content (Figure 4b). The proportion of sorbed copper on Al-bentonites decreases with an increase in the initial solution concentration from 40–50% to 25% and from 30% to 10% for Al$_{13}$ and HAl$_{13}$ respectively.

In the range of concentrations of the initial copper chloride solutions from 0.025 to 0.125 mmol/L, the equilibrium pH values in the experiments with Na-bentonite do not vary and exceed the pH value of the initial solution consistently by 1 unit. However, beyond this, the equilibrium pH decreases sharply and at 0.5 mmol/L, the pH value is higher than the initial only by 0.3 units (Figure 5).

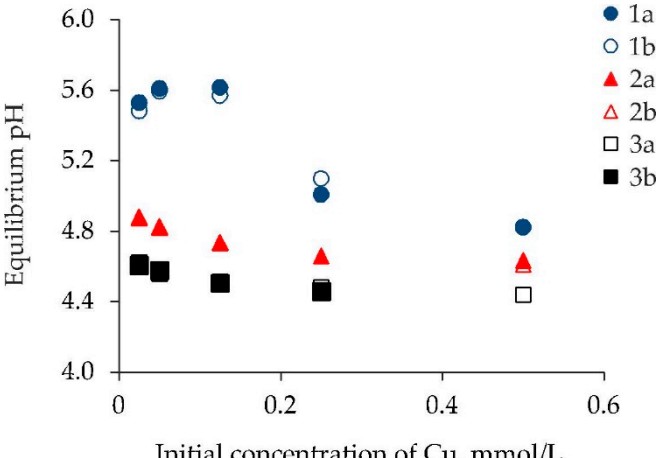

**Figure 5.** The dependence of the equilibrium pH value on the initial concentration of Cu(II). 1, 2, 3 = Na-, HAl$_{13}$- and Al$_{13}$-bentonite, respectively. Empty and filled icons—repetition of the experiment.

The opposite dependence is observed for Al-bentonites (Figure 5). In this case, an increase in pH compared to the initial value does not exceed 0.3 units. In general, HAl$_{13}$-bentonite equilibrium pH appeared consistently higher by 0.2 units compared with Al$_{13}$-bentonite.

### 3.4. Sorption of Humic Acid by Na- and Al$_{13}$-bentonites

It was found that the maximum amount of humic acid from the solution with concentration C$_{HA}$ = 23.4 mmol/L, I = 0.1 mmol/L and pH 4.5 is adsorbed in 6 h of interaction with the calcined Al$_{13}$-bentonites. This maximum corresponds to about 38% of the initial amount of humic acid in solution. A longer reaction time (8 h) leads to the desorption of HA (Figure 6).

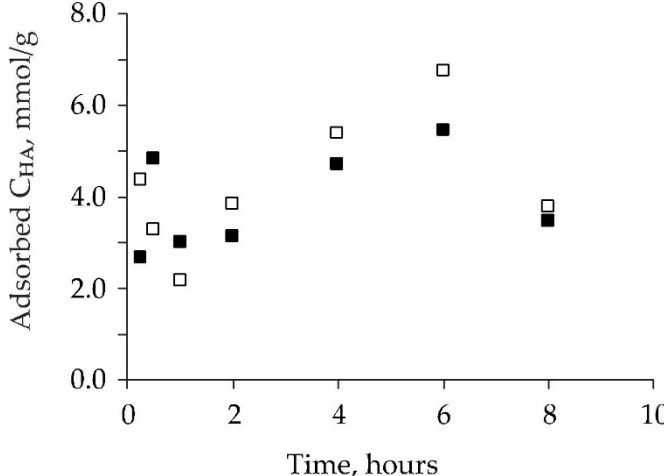

**Figure 6.** The dependence of the adsorbed amount of C$_{HA}$ on the time of interaction with Al$_{13}$-bentonite. Empty and filled icons—repetition of an experiment.

The sorption of HA on Na-bentonite from a solution with low concentration is virtually independent of the pH of the solution, while from higher concentration HA solutions the sorption is largely reduced with increasing pH (Figure 7a,b).

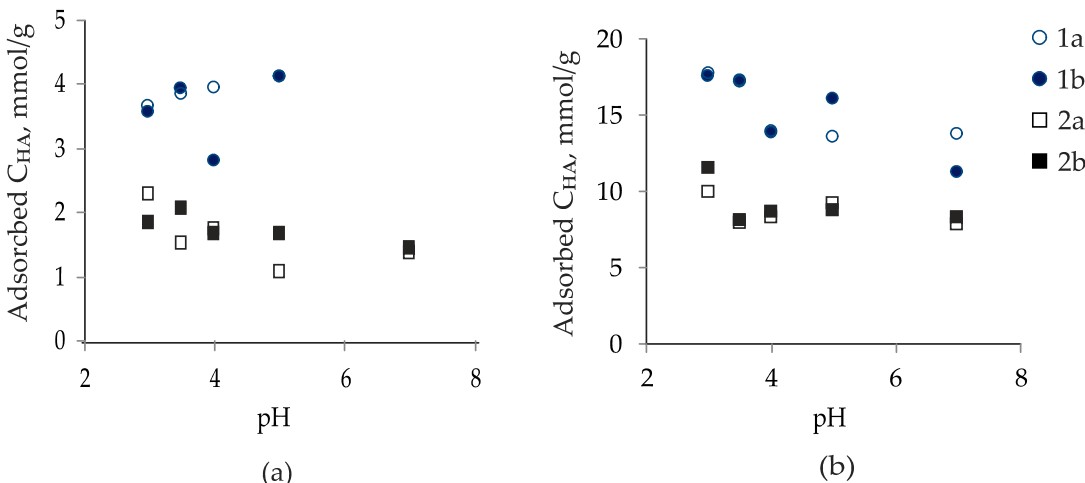

(a)

(b)

**Figure 7.** The dependence of the adsorbed HA amount from HA solutions with concentrations of 6.09 mmol/L (**a**) and 24.35 mmol/L (**b**) on pH values. I = 0.1 mol/L. 1, 2 = Na- and Al$_{13}$-bentonite, respectively. Empty and filled icons—repetitions of an experiment.

The sorption of humic acid on Al$_{13}$-bentonite increases slightly at pH < 3.5 and practically does not depend on pH in the range of pH 3.5–8 (Figure 7a,b).

Na-bentonite adsorbs more HA compared to Al$_{13}$-bentonite and the proportion of the sorbed HA does not change in the whole range of concentrations (Figure 8).

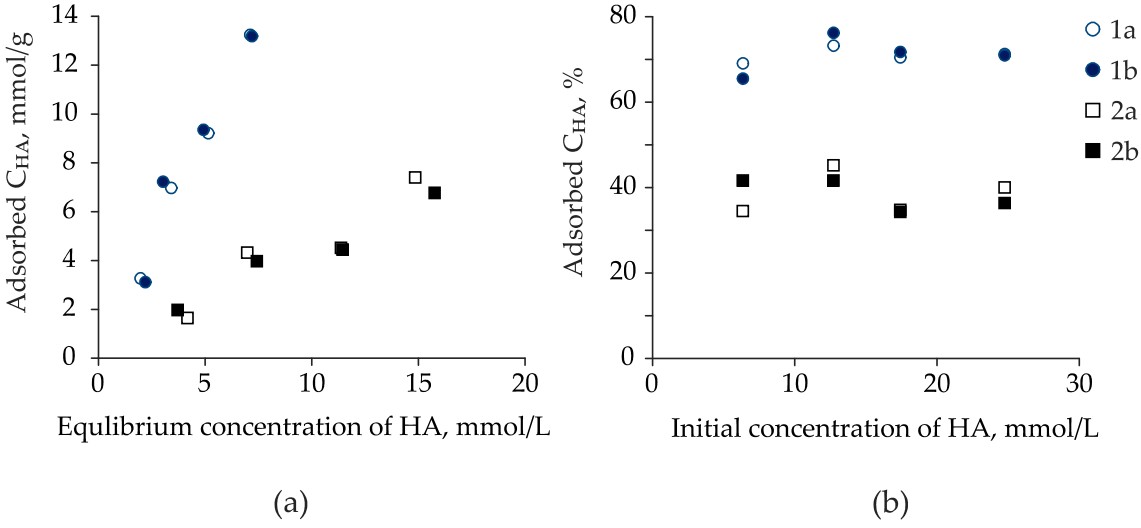

(a) (b)

**Figure 8.** The dependence of the sorption of Cu(II) in mmol/g (**a**) and % (**b**) on Na-bentonite (1) and Al$_{13}$-bentonite (2). Empty and filled icons (a,b)—repetitions of the same experiment.

### 3.5. Sorption of Cu$^{2+}$ Ions by HAAl$_{13}$-bentonite

The amount of sorbed Cu$^{2+}$ ions increases with their concentration in the initial and equilibrium solutions and the proportion of adsorbed ions decreases expectedly with increasing concentration of the initial copper chloride solution (Figure 9).

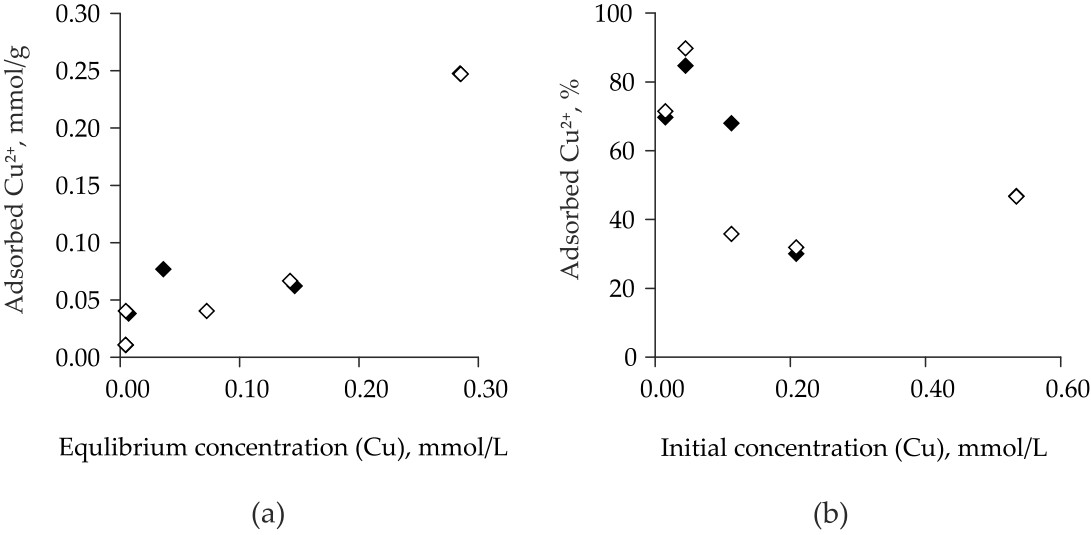

(a) (b)

**Figure 9.** The sorption isotherm (**a**) and the fraction of the adsorbed Cu$^{2+}$ ion (**b**) on the Al$_{13}$-bentonite HA (*1, 2*). Empty and filled icons—repetitions of an experiment.

The maximum amount of sorbed Cu$^{2+}$ ions on the HAAl$_{13}$-bentonite is 0.25 mmol/g and this value is close to the maximum quantity of sorbed Cu$^{2+}$ ions by Na-bentonite which equates to 0.33 mmol/g in the conditions of these experiments (Figures 4a and 9a). The proportion of the adsorbed Cu(II) ions on the HAAl$_{13}$-bentonite at initial concentrations of copper chloride <0.05 mmol/L varies from 60 to 90%, which is comparable with those for the Na-bentonite and significantly higher than those of the HAl$_{13}$ and Al$_{13}$ forms (Figures 4b and 9b)

## 4. The Discussion of the Results

### 4.1. Regularities of Sorption of $Cu^{2+}$ by Different Bentonite Forms

Calculations performed in the Visual MINTEQ program showed that the proportion of $Cu^{2+}$ ions in the solutions used for the sorption experiments is 0.99.

Bentonite saturated with $Na^+$ ions under experimental conditions, i.e., at pH 4.5 in the range of ionic strengths from $7.5 \times 10^{-5}$ to $1.5 \times 10^{-3}$ mol/L sorbs 2–6 times more $Cu^{2+}$ than $HAl_{13}$-and $Al_{13}$-bentonite (Figure 4). The values of the coefficient $k$ in the Freundlich equation, which satisfactorily approximates to obtained adsorption isotherms, are significantly higher for Na-bentonite than for $HAl_{13}$ and $Al_{13}$-bentonites (Table 1). There are two main mechanisms of sorption of cations by montmorillonite—ion exchange and adsorption on the surface [33–37]. Saturation of bentonite with Al Keggin cations resulted in a significant decrease of CEC from 104 cmol(+)/kg for Na-bentonite, to 29 and 39 cmol(+)/kg for $HAl_{13}$- and $Al_{13}$-bentonite respectively (Table 1).

**Table 1.** CEC and the parameters of the Freundlich equation for different forms of bentonite.

| Bentonite | CEC, cmol(+)/kg | Sorbate | $k$ | $n$ | $R^2$ |
|---|---|---|---|---|---|
| Na | 104 | $Cu^{2+}$ | 4.9 | 1.0 | 0.90 |
| | | HA | 0.2 | 1.0 | 0.95 |
| $HAl_{13}$ | 29 | $Cu^{2+}$ | 0.1 | 0.6 | 0.94 |
| $Al_{13}$ | 39 | $Cu^{2+}$ | 0.2 | 0.8 | 0.97 |
| | | HA | 0.002 | 1.0 | 0.93 |
| $HAAl_{13}$ | Undefined | $Cu^{2+}$ | 0.3 | 0.5 | 0.76 |

Given the fact that, in the experiments at the initial concentration of $CuCl_2 \leq 0.125$ mmol/L, the pH of the equilibrium solution remains constant (Figure 5), it can be assumed that the main mechanism of sorption of $Cu^{2+}$ ions on Na-bentonite in the given range of concentration is cation exchange.

When using more concentrated solutions of copper chloride, an insignificant decrease of pH by 0.5–0.7 units is observed, which is probably caused by the substitution of $H^+$ in aluminol groups by $Cu^{2+}$ ion and the formation of surface complexes on the side edges of montmorillonite. It was shown that the pH of the point of zero net proton charge ($pH_{PZC}$) of the montmorillonite edge surfaces is about 3.4 units [38]. According to other authors, the $pH_{PZC}$ of montmorillonite varies between 4.7–7.8 [39–42]. According to calculations conducted in [34] the pKa value of silanol $(\equiv Si - O- \Leftrightarrow \equiv SiOH)$ and aluminol $(\equiv Al(OH)2 \Leftrightarrow \equiv OHOH2)$ groups of montmorillonites are 7.0 and 8.3, respectively.

Taking the above into account, we can conclude that the pH-dependent edge-sites on montmorillonites are sufficiently protonated in the conditions of the experiment and do not make significant contributions to the adsorption of $Cu^{2+}$ ions. The value of the parameter $n$ in the Freundlich equation for Na-bentonite is *1* which might be evidence of the homogeneity of sorption sites.

Therefore, it can be concluded that in the investigated range of concentrations and at a pH of 4.5 of the initial copper chloride solutions, the cation exchange reaction $Cu^{2+} \leftrightarrow Na^+$ is the main mechanism of $Cu^{2+}$ sorption on Na-bentonite

The saturation of bentonite with Al Keggin cations leads to a decrease of CEC (Table 1) and to an increase in the number and proportion of pH-dependent sorption sites due to the "pillar" structures in montmorillonite interlayers. The titration curves of Na-bentonite obtained using a solution of sodium chloride of different concentrations do not intersect, which indicates the predominance of a constant charge (Figure 10a). The predominance of the pH-dependent charge in $Al_{13}$-bentonite ensures the intersection of the titration curves at the point corresponding to $pH_{PZC} = 4.2$ (Figure 10b). The obtained $pH_{PZC}$ values for $Al_{13}$-bentonite are consistent with published data [26].

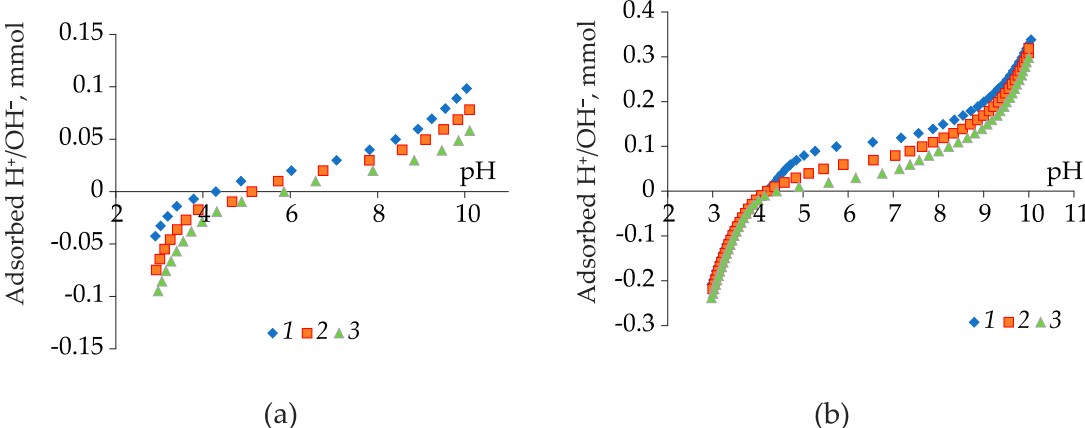

**Figure 10.** Dependence of the amount of adsorbed H$^+$ or OH$^-$ on the pH of Na- (**a**) and Al$_{13}$-bentonite (**b**). NaCl concentration: 0.1 M (*1*), 0.01 M (*2*) and 0.001 M (*3*).

Given the fact that in the experiments with Al$_{13}$-bentonite the equilibrium pH values are equal to or higher than the pH$_{PZC}$ by less than 0.2–0.4 units, it can be concluded that a significant proportion of the reaction surface of Al$_{13}$-bentonite is neutrally charged or has a positive charge, which complicates the sorption of Cu$^{2+}$ ions on these surfaces from a solution of copper chloride with a pH 4.5. Thus, the saturation of bentonite with Al Keggin cations leads to a significant decrease in CEC, and the additionally formed pH-dependent adsorption sites, for the mentioned reasons, provide much lower sorption of Cu$^{2+}$ ions in comparison with ion-exchange reactions. The pH$_{PZC}$ of HAl$_{13}$-bentonite was not determined in this work, however, it can be assumed that its value would not differ significantly from the pH$_{PZC}$ of Al$_{13}$-bentonite, which would explain the deterioration of the sorption characteristics of HAl$_{13}$-bentonite in comparison with Na-bentonite towards copper ions. Similar results were obtained in experiments on the sorption of Cu(II) ions on Al$_{13}$-bentonite at pH 4.9 [16,43].

A reduction in the equilibrium pH values due to the sorption of Cu$^{2+}$ ions on HAl$_{13}$- and Al$_{13}$-bentonite in the initial copper chloride solution concentration range from 0.025 to 0.25 mol/L (Figure 5) can be explained by deprotonation of aluminol groups and the formation of inner-sphere surface complexes. The presence of the dependence of the pH value on the concentration of the initial copper chloride solution indicates that the sorption of Cu$^{2+}$ ions on Al-bentonite is mainly carried out by the formation of surface complexes. For both forms of Al-bentonite, the Freundlich parameter *n* turned out to be < 1, which can be explained by the heterogeneity of the sorption centers, which is more pronounced in the case of HAl$_{13}$ bentonite (Table 1).

## 4.2. Regularities of HA Sorption on Na- and Al$_{13}$-bentonites

Humic acids can be adsorbed on montmorillonite by ligand exchange, electrostatic interactions, hydrogen bonding, van der Waals and hydrophobic interactions. Sorption of humic acids on the montmorillonite surface depends on several factors, including the pH, the structure and concentration of the HA, the ionic strength of the solution, and the interaction [44,45]. Both external basal and edge surfaces of montmorillonite are available for the sorption of humic acids.

Siloxane surfaces are mainly hydrophobic (in parts where there is no charge effect and accordingly no influence of the charge-compensating cations) and the edge surfaces have hydrophilic properties due to silanol and aluminol groups. Therefore, the sorption of HA on the basal surfaces can be carried out by hydrophobic interactions. However, sorption by this mechanism does not depend on pH. At montmorillonite's basal surfaces, a constant negative charge is concentrated and therefore compensated for by Na$^+$ cations. Since the Na$^+$ cations are weakly bonded to the montmorillonite surface in the form of outer-sphere complexes, it can be assumed that at low pH there is a possibility for competition between protonated functional groups of HA and Na$^+$ ions for the sorption sites on the external basal surfaces, which leads to a significant increase of sorption with decreasing pH. An

increase in pH is accompanied by deprotonation of the functional groups of HA and, as a consequence, to the repulsion of HA from the negatively-charged montmorillonite basal surfaces.

At pH < 4.2 silanol and aluminol groups on the edges of montmorillonite are mainly in a protonated state and do not take part in the sorption of HA. Increasing the pH results in the deprotonation of these functional groups and a stronger repulsion of HA from the edge surfaces of montmorillonite.

It is shown that the adsorption of organic compounds involving van der Waals interactions and hydrogen bonding is virtually pH-independent [45].

Based on the above, it can be assumed that the main mechanism of HA sorption from a 24.35 mmol/L solution on Na-bentonite (Figure 7b) is electrostatic interaction with external basal surfaces. Sorption of HA on Na-bentonite from solutions with concentrations $C_{HA}$ from 6.09 to 24.35 mmol/L at pH 3 and I = 0.1 mol/L is satisfactorily approximated by the Freundlich equation and a value of $n = 1$ may indicate the uniformity of sorption sites (Table 1).

$Al_{13}$-bentonite adsorbs much less humic acid compared to Na-bentonite (Figure 8a,b). It can be explained by the fact that after the saturation of Na-bentonite with AL Keggin cations a significant proportion of negative charge is compensated by a highly charged cation $[AlO_4Al_{12}(OH)_{24}(H_2O)_{12}]^{7+}$, which is strongly bonded on the surface and in the interlayer of montmorillonite and probably only a small amount might be substituted with protonated functional groups of HA with decreasing pH. Consequently, the possibility of HA sorption due to electrostatic interaction on the basal surfaces of $Al_{13}$ bentonite decreases. A slight increase in HA sorption by $Al_{13}$-bentonite at pH < 3.5 can be explained by the incomplete substitution of exchangeable cations with $Na^+$ during the preparation of Na-bentonite. In the range of pH from 3.5 to 7, HA adsorption on the $Al_{13}$-bentonite occurs to a lesser extent than on the Na-bentonite and does not depend on pH. Artificially created additional pH-dependent sites on the surface of $Al_{13}$-bentonite in highly acidic conditions are becoming protonated and have a positive charge, while at pH > 4.2 they become deprotonated, and the proportion of their negative charge increases. Therefore, the electrostatic interactions between the functional groups of the HA and the montmorillonite surface under these conditions are most likely to be reduced. The lack of pH dependence of the sorption at pH > 3.5 indicates that the main mechanisms of HA sorption may be hydrophobic, van der Waals interactions and hydrogen bonding that are essentially independent of pH.

The value of the parameter $n = 1$ in the Freundlich equation describing the sorption of HA on $Al_{13}$-bentonite in the studied concentration range at pH 3 and I = 0.1 mol/L may indicate the homogeneity of the sorption centers (Table 1).

The lack of a dependence, or the presence of a weak dependence, of the amount of sorbed HA of pH at $C_{HA}$ = 6.09 mmol/L (Figure 7a) can be explained by small concentrations of $C_{HA}$ and errors in a determination by the spectroscopic method.

In addition to the considered mechanisms of HA sorption on different forms of bentonite, coagulation at low pH values cannot be excluded. However, this process has not been studied in this present work.

### 4.3. Sorption of $Cu^{2+}$ Ions by $HAAl_{13}$-bentonite

The sorption of $Cu^{2+}$ ions by $HAAl_{13}$-bentonite in the range of studied concentrations is described by the Freundlich equation, although with a lower coefficient of determination in comparison with Na and Al forms of bentonite (Table 1). An increase in sorption and in the degree of extraction of $Cu^{2+}$ ions from solutions by $HAAl_{13}$-bentonite in comparison with $HAl_{13}$- and $Al_{13}$-bentonite can be explained by the formation of additional sorption sites on the surface of humic acid-modified $Al_{13}$-bentonite. It was shown that the sorption of humic acids on Na-bentonite, Ca-bentonite and goethite leads to a significant increase in the amount of adsorbed metal cations by increasing the amount of the sorption sites on the surface of minerals available for complexation [21,46,47]. The value $n = 0.5$ in the Freundlich equation probably indicates the inhomogeneity of sorption sites on the obtained organomineral complex.

As already mentioned above, the sorption experiments of $Cu^{2+}$ ions by $HAAl_{13}$-bentonite was carried out at constant ionic strength of 0.1 mol/L, and in experiments, with Na- and $Al_{13}$-bentonites ionic strength ranged between $10^{-5}$ up to $10^{-3}$ mol/L. It is known that an increase of ionic strength leads to a decrease of sorption of both metal ions and humic acid [39,40,48–50]. However, a change in the ionic strength in the range from $10^{-5}$ to $10^{-3}$ mol/L affects the sorption of $Cu^{2+}$ ions slightly. With an increase in ionic strength to 0.1 mol/L, the amount of sorbed metal ions decreases significantly [51]. Despite the fact that in the sorption experiments of $Cu^{2+}$ by $HAAl_{13}$-bentonite, the ionic strength was in 2–4 times higher than that in the experiments with Na-bentonite, the amount of sorbed $Cu^{2+}$ ions and the degree of extraction from solutions are similar to those for Na-bentonite

## 5. Conclusions

1. The modification of Na-bentonite with Al Keggin cations leads to an increase in the interplanar spacing of montmorillonite from 1.29 nm for the Na-form to 1.85 and 1.78 nm for $HAl_{13}$- and $Al_{13}$-forms respectively; a reduction of CEC; and the formation of additional surfaces with a variable charge having $pH_{PZC}$ 4.2. $Al_{13}$-bentonite is not affected by heat.

2. Under the conditions of the experiments carried out, Na-bentonite adsorbs more $Cu^{2+}$ ions from $CuCl_2$ solutions with a pH of 4.5, compared to $Al_{13}$ forms of bentonites. The main mechanism of copper sorption on Na-bentonite is the cation exchange $Cu^{2+} \leftrightarrow Na^+$. The reduction of CEC of Na-bentonite after modification with Al Keggin cations leads to a decrease in the sorption capacity of $Cu^{2+}$. pH-dependent sorption sites formed on $Al_{13}$-bentonites have a $pH_{PZC}$ of 4.2 and, therefore, under the conditions of the experiment carried out, have a significant positive charge and prevent intensive sorption of $Cu^{2+}$ ions. Sorption of $Cu^{2+}$ ions by all studied bentonite forms can be satisfactorily approximated by the Freundlich equation.

3. Na-bentonite adsorbs more HA, compared with $Al_{13}$-bentonite and the proportion of adsorbed HA does not change over the entire concentration range. A decrease in pH leads to an increase in the sorption of HA by Na-bentonite and weakly affects the HA sorption by $Al_{13}$-bentonite.

The main sorption mechanism of HA by Na-bentonite is electrostatic interaction with the external basal surfaces, while for $Al_{13}$-bentonite it is a combination of hydrophobic, van der Waals interactions and hydrogen bonding that are virtually independent of pH. At low pH values, the coagulation of HA is possible and may affect the obtained results.

4. Treatment of the $Al_{13}$-bentonite surface with HA leads to the formation of the additional sorption sites. The amount of sorbed $Cu^{2+}$ ions and the degree of their extraction from solutions by $HAAl_{13}$-bentonite is close to those values for Na-bentonite.

**Author Contributions:** Conceptualization, methodology I.T. and Y.I.; data curation, formal analysis, and investigation, Y.I., I.G., M.K. and S.Z.; writing, I.T. and V.K. All authors have read and agreed to the published version of the manuscript.

**Funding:** Experiments on sorption of copper and HA were carried out by the state budget topic of chemistry department, Soil Science Faculty, MSU (Topic N° 116011910186-5). Investigations bentonite modification were founded by the Russian Foundation for Basic Research (grant N° 18-29-12115).

**Conflicts of Interest:** The authors declare no conflict of interest.

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
