# Peer review of "Sorption of Cu2+ Ions by Bentonite Modified with Al Keggin Cations and Humic Acid in Solutions with pH 4.5"

_minerals, doi:10.3390/min10121121_

Round 1

Reviewer 1 Report

The manuscript is focused on the study of the sorption process of Cu2+ ions by bentonite modified both Al Keggin cations and humic acid,

The manuscript is in the topics of the Minerals, concisely written with logical links throughout the whole text. I appreciate the amount of sorption experiments. The data worth publishing. Regardless, the manuscript needs minor revision.

Pay attention to SI units and give all dimensions according to the standard style of the Journal with special attention to units for Figures 2,3,6

Fig. 1 - The axis "y" labeled as Intensity without a unit is recommended

I would recommend using the appropriate moduls for the creation of equations (e.g. link 133) and formulas (link 300)

Also correct please some typing errors like e.g.: line 15 - CuCl2, line 137 - measurements etc.

Author Response

On behalf of the co-authors, I would like to thank the reviewer for his work and comments.

Below, answers to comments point by point.

- Pay attention to SI units and give all dimensions according to the standard style of the Journal with special attention to units for Figures 2,3,6

Answer: Thank you for these notes, figures have been corrected.

- Fig. 1 - The axis "y" labeled as Intensity without a unit is recommended

Answer: Thank you for this comment, figure with xrd patterns corrected, names of axis added.

- I would recommend using the appropriate moduls for the creation of equations (e.g. link 133) and formulas (link 300)

Answer: Thank you for the comments, formulas were corrected.

- Also correct please some typing errors like e.g.: line 15 - CuCl2, line 137 - measurements etc.

Answer: Thank you for the comments, it has been corrected.

Reviewer 2 Report

Remarks

  • “ The sorption of Cu2+ onto bentonite modified with Al Keggin cations and humic acid from CuCl2 solutions at pH 4.5 was studded”.

The justification (reason) for the choice of bentonite modification by Al Keggin cations as well as the pH value 4.5 should be added in the manuscript.

  • Did the authors expect an increase in copper absorption when the modification of Na-bentonite with Al Keggin cations caused a decrease in CEC value of bentonite?

  • Materials and Methods
  1. “The pH of the clay fraction suspension in Ca- and Na- forms were measured at a clay: water ratio of 1 : 1000 and was estimated at 6.21 and 6.38 units respectively”.

Is it possible to obtain  the clay fraction suspension at a clay: water ratio of 1 : 1000?

  1. “The experiments were carried out at an ionic strength of 0.1 M, which was maintained using NaCl solution”.

Why so high ionic strength of 0.1 M NaCl  (5,84 g/L) was used in the experiment?

  1. “The dependence of the HA sorption on pH was studied using 24.35 mM and 09 M solutions...”.

A humic acid molecular weight is about 230 g/mol. So, is it not too high the concentration 24.35 mM (about 6 g/l)?

  1. 4. Meuseremne „The content of Corg in the initial HA solution was determined by the Tyurin method with the photometric ending [21]. The Corg concentration in the equilibrium solutions was determined by spectrophotometry using a UNICO 1201 spectrophotometer (United Products & Instruments Inc., USA)”.

Is it possible to determine Corg concentration in solutions by spectrophotometry method at 350 nm wavelength?

  • 2. Kinetics of Cu2+ adsorption
  1. Please correct the sorption unit on fig 2a
  2. Why the results of kinetics of Cu2+ ions adsorption by HAAl13-bentonite are absent?

  • 4. Sorption of humic acid by Na- and Al13-bentonites.

According to study research data (fig. 7b) it was found that the maximum amount of humic acid from the solution with concentration CHA  = 23.4 mmol/L is about  17 mmol/g) or about 4 g/g (MW of HA 230 g/mol. Is it true?

Author Response

On behalf of the co-authors of the article, I would like to thank the reviewer for the time spent and the comments made, which made it possible to make the article better.
Below, I will answer the comments point by point.

  • “ The sorption of Cu2+ onto bentonite modified with Al Keggin cations and humic acid from CuCl2 solutions at pH 4.5 was studded”.

The justification (reason) for the choice of bentonite modification by Al Keggin cations as well as the pH value 4.5 should be added in the manuscript.

Answer: In an acidic environment, the sorption of cations on the surface of bentonite treated with humic acid will be limited by the pKa values of the carboxyl groups of humic acid, which are known to vary in the range 5 –  7.5. It is necessary to establish the efficiency of bentonite modified in this way for the sorption of cations in the acidic environment.

These text has been added to the article.

  • Did the authors expect an increase in copper absorption when the modification of Na-bentonite with Al Keggin cations caused a decrease in CEC value of bentonite?

Answer: Bentonite modified with Keggin cations of aluminum after calcination becomes thermally stable and swells to a lesser extent, which makes it possible to use it as a sorbent or catalyst under certain environmental conditions. It is clear that such a modification causes the decrease in the CEC to the respective the reduction in the sorption of cations. The ability of such bentonite to adsorb cations can be increased by modifying its surface. In our work, we showed that humic acid can be applied as a surface modifier.

  • Materials and Methods

“The pH of the clay fraction suspension in Ca- and Na- forms were measured at a clay: water ratio of 1 : 1000 and was estimated at 6.21 and 6.38 units respectively”.

Is it possible to obtain  the clay fraction suspension at a clay: water ratio of 1 : 1000?

Answer: It is possible.

  • “The experiments were carried out at an ionic strength of 0.1 M, which was maintained using NaCl solution”.

Why so high ionic strength of 0.1 M NaCl  (5,84 g/L) was used in the experiment?

Answer: It is true, the ionic strength is actually high but it is normal for similar experiments and is using very often.

  • “The dependence of the HA sorption on pH was studied using 24.35 mM and 09 M solutions...”.

A humic acid molecular weight is about 230 g/mol. So, is it not too high the concentration 24.35 mM (about 6 g/l)?

Answer: 24.35 mM is not the concentration of humic acid, but it is the C concentration of humic acid. Explanations are added to the text.

  • Measurement „The content of Corg in the initial HA solution was determined by the Tyurin method with the photometric ending [21]. The Corg concentration in the equilibrium solutions was determined by spectrophotometry using a UNICO 1201 spectrophotometer (United Products & Instruments Inc., USA)”.

Is it possible to determine Corg concentration in solutions by spectrophotometry method at 350 nm wavelength?

The Corg concentration in the initial HA solution was determined by spectrophotometry at 590 nm after the oxidation of organic matter with K2Cr2O7 (Tyurin's method). The respective additions have been made to the text.

Followed text has been added: The content of Corg in the initial HA solution was determined by the Tyurin method with the photometric ending at 590 nm  wavelength. 

 The concentration of Corg in equilibrium solutions was determined spectrophotometrically at a wavelength of 350 nm, since the absorption maxima at this wavelength correlated well with the DOC concentration. A similar approach to determining the concentration of DOC in waters was used in the works listed below at wavelengths 365, 290 - 350, and 350 - 500 nm..

 Journal of Water Management and Research 72:169–175. Lund 2016

HYDROLOGICAL PROCESSES Hydrol. Process. 21, 3181–3189 (2007) Published online 25 September 2007 in Wiley InterScience (www.interscience.wiley.com) DOI: 10.1002/hyp.6887

A. Ghabbour & G. Davies, Annals of Environmental Science / 2009, Vol 3, 131-138

Follow text has been added: UNICO 1201 spectrophotometer (United Products & Instruments Inc., USA), since the absorption maxima at this wavelength correlated well with the DOC concentration.

  • 2. Kinetics of Cu2+ adsorptio
  • Please correct the sorption unit on fig 2a

Answer: Thank you for comment. it has been corrected.

  • Why the results of kinetics of Cu2+ ions adsorption by HAAl13-bentonite are absent?

Answer: Experiments on the kinetics of sorption of Cu2 + HAAl13-bentonite were not performed.

  • 4. Sorption of humic acid by Na- and Al13-bentonites.

According to study research data (fig. 7b) it was found that the maximum amount of humic acid from the solution with concentration CHA  = 23.4 mmol/L is about  17 mmol/g) or about 4 g/g (MW of HA 230 g/mol. Is it true?

24.35 mM is not the concentration of humic acid, but it is the C concentration of humic acid. Explanations are added to the text.

Follow text has been change to: The dependence of the HA sorption on the time of interaction was studied using solution with concentration CHA  = 23.4 mmol/L Сorg 24.35 mM  at pH 4.5 with a clay:...

On behalf of the co-authors of the article, I would like to thanks again!

Sincerely,

Victoria Krupskaya

Reviewer 3 Report

The introduction must be expanded as it is very limited. There is a lot of work published on bentonite and the removal of heavy metals, see for instance in https://doi.org/10.1080/19443994.2016.1235153. 

Why the specific modifications were tested?

Section 2.3 - how the authors ensured that equilibrium was attained? 

In Figure 2 onwards instead of repetitions please use the average values and the standard deviation/error bars

Author Response

On behalf of the co-authors, I would like to express my thanks for the reviewing of our article and the comments made.
Below, answers to remarks point by point.

- The introduction must be expanded as it is very limited. There is a lot of work published on bentonite and the removal of heavy metals, see for instance in https://doi.org/10.1080/19443994.2016.1235153

Answer: Authors thanks Reviewer for this comment. Introduction has been expanded, advised reference has been added.

Follow text has been added:

The main mechanisms of sorption of Cu (II) ions on bentonite are ion exchange and proton substitution of aluminol and silanol groups on the edge surfaces of clay minerals. Ion exchange is independent of pH. Sorption on aluminol and silanol groups depends on pH

- Why the specific modifications were tested?

Answer: After modification with Keggin cations and calcination bentonite becomes stable to heating and its swelling reduces, which makes it possible to use it as an effective sorbent and catalyst under certain conditions. However, the modification causes the decrease in CEC. The adsorptive capacity of the Al13-bentonite can be improved by treating its surface with humic acid.

This test has been added.

- Section 2.3 - how the authors ensured that equilibrium was attained?

Answer: Before carrying out the main experiment, we estimated the dependence of the copper ions sorption upon time. It was found, that already 20 minutes after the start of the experiment, the sorption of copper did not change significantly (Fig. 2). Therefore, it was accepted that 6 hours is quite enough for equilibrium or “quasi-equilibrium” state to be established in the system.

- In Figure 2 onwards instead of repetitions please use the average values and the standard deviation/error bars

Answer: The experiments were carried out in duplicate which is not enough for statistical processing, and this is why no error bars are represented on the plots. To show the possible variability of the results related to random errors at all stages of the experiment the data for each of two replicates are given on the plots.

We would like to thanks again!

Victoria Krupskaya and co-authors.